# Resilience and Psychological Well-Being of Polish Women in the Perinatal Period during the COVID-19 Pandemic

**DOI:** 10.3390/jcm12196279

**Published:** 2023-09-29

**Authors:** Klaudia Sójta, Aleksandra Margulska, Maksymilian Plewka, Kacper Płeska, Dominik Strzelecki, Oliwia Gawlik-Kotelnicka

**Affiliations:** 1Department of Affective and Psychotic Disorders, Medical University of Lodz, Czechoslowacka Street 8/10, 92-216 Lodz, Poland; klaudia.krakus@stud.umed.lodz.pl (K.S.); dominik.strzelecki@umed.lodz.pl (D.S.); 2Department of Adolescent Psychiatry, Medical University of Lodz, Czechoslowacka Street 8/10, 92-216 Lodz, Poland; aleksandra.margulska@umed.lodz.pl; 3Faculty of Medicine, Medical University of Lodz, al. Kosciuszki 4, 90-419 Lodz, Poland; maksymilian.plewka@stud.umed.lodz.pl (M.P.); kacper.pleska@stud.umed.lodz.pl (K.P.)

**Keywords:** COVID-19 pandemic, perinatal mental health, resilience, depression, anxiety related to the childbirth

## Abstract

Purpose: The COVID-19 pandemic, with its multidimensional consequences, is the most serious threat of the 21st century affecting the mental health of women in the perinatal period around the world. Resilience, which assumes the flexible use of an individual’s resources in facing adversity, is an important, protective factor influencing mental well-being. The presented study aimed to determine to what extent psychological resilience, mitigates the relationship between adverse consequences of the COVID-19 pandemic and symptoms of depression and anxiety in women in the perinatal period. Methods: We recruited pregnant women from 17 February to 13 October 2021, using social media, the parenting portal, and the snowball method. To assess mental well-being, we used: The Edinburgh Postnatal Depression Scale (EPDS), The Beck Depression Inventory (BDI-2), Self-report Labour Anxiety Questionnaire—LAQ and the self-developed COVID-19 Pandemic Anxiety Questionnaire (CRAQ). Resilience was measured usingthe Resilience Measure Questionnaire (KOP26). Multiple Correspondence Analysis (MCA), an independent t-test, and a Pearson correlation analysis were performed. Results: Low resilience was significantly associated with depressive symptoms (r = −0.46; *p* < 0.05) and anxiety related to childbirth (r = −0.21; *p* < 0.05). No associations were found for resilience and pandemic-related stress. Very high and high perinatal anxiety along with the lowest level of resilience clustered with EPDS and BDI-2 scores indicating depression. Conclusions: Our study provides evidence that lower levels of resilience during pregnancy may be a significant predictor of increased severity of depressive symptoms and higher levels of anxiety related to childbirth among the perinatal population.

## 1. Introduction

The perinatal period is characterized by a variety of essential changes and great challenges. The presence of strong, often opposing emotions accompanying pregnancy and early parenthood makes this time particularly burdensome for mental well-being [1,2]. Depressive disorders affect women two to three times more often than men during their lifetime, and the reproductive period poses particular risks to their development [3]. Previous research has shown that anxiety is even more prevalent than depression during pregnancy and often co-occurs with perinatal depression [4,5]. The prevalence of major to minor depressive disorders ranged from 3.1% to 11% in the antenatal period and from 4.7% to 13% in the first 3 months after delivery, respectively [6]. In international studies, the overall prevalence of any anxiety disorder during pregnancy is estimated at 15.2% and 9.9% after birth (1–24 weeks postpartum) [7]. These estimates relate mainly to high-income countries, and higher prevalence rates of perinatal depression (regardless of episode severity) and anxiety are generally reported in women in low- and middle-income countries [6,7]. Perinatal mental disorders have been reported to be associated with negative health outcomes for both mother and infant [8,9]. Women with perinatal depression and anxiety are more likely to receive poorer social support, report significantly lower quality of life, and are at an increased risk for substance use, work-related dysfunction, malnutrition, hypertension, and other obstetric complications, such as pre-eclampsia, caesarean section and episiotomy [10,11,12,13,14]. There is also a negative relationship between prenatal attachment and perinatal depression and anxiety [15,16]. Untreated perinatal depression and anxiety affect the child’s health and development, and these outcomes are both short-term (e.g., premature birth and low birth weight, high incidence of infectious diseases) and long-term (e.g., cognitive, emotional, motor, and neural developmental deficits observed until adolescence) [10,17,18,19,20,21,22].

In December 2019, the first cases of severe pneumonia, caused by the novel coronavirus were reported in Wuhan, China’s Hubei province [23]. On 30 January 2020, the WHO issued a Public Health Emergency of International Concern (PHEIC) as a result of the SARS-CoV-2 epidemic, and later, on 11 March, declared the COVID-19 outbreak a global pandemic [24]. Despite the positive impact on the rate and range of COVID-19 transmission, quarantine, and social isolation disturbed the psychosocial well-being of the individuals [25]. Subsequent studies have consistently reported that the COVID-19 pandemic has caused serious mental health problems worldwide, comparable to the consequences of major disasters and armed conflicts [26].

Over the past years, the priority of perinatal mental health has been growing, bringing new conclusions and solutions [27]. The pandemic period posed a particularly serious threat affecting the mental health of pregnant women internationally [28]. Additional COVID-19 precautions were taken during and after childbirth to protect the health of newborns, mothers, and healthcare workers. Apart from the protective effect, the restrictions related to the pandemic posed a threat to the psychological well-being of perinatal women. One of the most important changes in perinatal care were restrictions on labour companionship. Labour companion is a well-recognized support for women that affects the successful course of childbirth and provides health benefits for both the newborn and mother [29]. In Poland, the number of physiological births attended by a close person dropped from over 80% in 2018 to approximately 38% during the pandemic [30]. Other important issues regarding changes in perinatal care in Poland due to the pandemic are presented in Table 1 [31]. These restrictions, although slightly different depending on the country, have resulted in increasing stress and uncertainty for entire families, as exemplified by the nationwide protests in Ireland against restrictions in maternity hospitals [32].

The possible impact of the lockdown, with its deleterious consequences, has directed the researchers’ special attention to this high-risk population. The first of the systematic reports showed a statistically significant increase in the level of anxiety experienced by women in the perinatal period during the COVID-19 pandemic, while the increase in depressive symptoms did not reach a statistically significant level compared to the period without the pandemic [33]. A recent systematic review of perinatal women’s mental health during the COVID pandemic found clinically significant symptoms of depression in 9.9% to 49% of participants in selected studies and clinically significant symptoms of anxiety in 11% to 61% of participants in selected studies. In addition, in a separate study, 10.2% of participants experienced PTSD symptoms. Other mental health outcomes indicated by the researchers include significant stress symptoms, feelings of loneliness, irritability, and increased worrying [32].

Contrary to known risk factors, some psychosocial resilience factors can buffer women in the perinatal period from negative consequences of distress. Resilience was originally understood as a multidimensional aspect of personality, ensuring the ability for healthy and positive adaptation during and after exposure to a highly adverse event [34]. However, because previous research failed to identify a single predictor of mental health in populations exposed to stress, the conceptualization of resilience was changed, defining it as the result of a complex and dynamic process of adaptation to adversity, resulting in the maintenance of mental well-being [35]. Resilience is considered both a mediator and moderator between stress exposure and symptoms of anxiety and depression [36,37,38]. Since the pandemic, there has been increasing interest in the impact of resilience on the ability to maintain mental health. Previous findings regarding the relationship between resilience and depressiveness in the perinatal period have been contradictory. The results of Kishore et al. suggested that resilience alone is not a strong enough factor to protect women from perinatal depression [39]. On the contrary, in the Chinese population, resilience was found to be a protective factor for maternal mental health, being both a mediator and moderator of the relationship between stress and prenatal anxiety and depression [40]. Similarly, the latest findings confirm that resilience has also been considered an important, protective factor influencing mental health in the perinatal periods experienced during the pandemic. Higher levels of resilience in peripartum women were associated with lower symptoms of depression and anxiety and lower overall concerns about COVID-19 [32]. Several resilience factors have been identified as the most protective in terms of levels of mental distress, which include social support, self-efficacy, adaptive emotion regulation strategies, regulatory flexibility, and optimism [35].

Therefore, the presented study aimed to determine to what extent psychological resilience factors, mitigate the relationship between adverse consequences of the COVID-19 pandemic and symptoms of depression and anxiety in women in the perinatal period.

## 2. Materials and Methods

### 2.1. Study Population and Methods

We recruited pregnant women from 17 February to 13 October 2021 (which covered the third and fourth wave of the COVID-19 pandemic in Poland), using social media, parenting portals, the snowball method, and with cooperation with the Department of Obstetrics and Gynecology of the Medical University of Lodz. Due to the popularization of vaccinations and their mitigating impact on the pandemic restrictions in maternity care, as well as the implementation of increasingly effective organizational procedures in maternity wards, the return to family deliveries or skin-to-skin contact between father and newborn after a cesarean section was enabled, therefore we decided to end recruitment for the study. Participation in this study was voluntary. To be eligible, participants must have either been pregnant or be in the postpartum period (up to six weeks) upon entering the study, living in Poland, and be fluent in Polish in order to be able to answer the questionnaires and be over 18 years old. We administered the questionnaires through Google Forms and set up a dedicated fanpage (https://www.facebook.com/BadanieKobiet/, accessed on 24 August 2023). All participants gave signed or provided electronically, informed consent to participate in the study. In the case of a score above the cut-off point for depressive symptoms (13 or more in EPDS), we sent participants an e-mail informing them about the occurrence of symptoms that may indicate a depressive episode, which requires additional assessment by a psychologist or psychiatrist and may require treatment. We drew attention to the need to inform the midwife or gynecologist about the result and have included a list of places offering support in the diagnosis and treatment of perinatal mental disorders. We obtained ethical approval by the Bioethics Committee at the Medical University of Lodz on 9 February (RNN/39/21/KE) for this research project. This cross-sectional study was part of a longitudinal examination to determine the relationship between resilience and mental health in perinatal women during the COVID-19 pandemic.

### 2.2. Measures

#### 2.2.1. Maternal Depression Symptoms

Two scales were used to assess the severity of depressive symptoms among women in the perinatal period: The Edinburgh Postnatal Depression Scale (EPDS) [41] and The Beck Depression Inventory (BDI-2) [42]. The EPDS is a 10-item self-report instrument rated on a 4-point Likert scale from 0 to 3 (total range of scores from 0 to 30). According to the recommendation of the National Institute for Health and Care Excellence (NICE), the EPDS is the first-choice screening test for perinatal depression [43]. This reliable tool is internationally validated and widely used. The most commonly adopted cut-off values used to identify women who might have depression are 11 or higher and 13 or higher [44]. The second most frequently used screening tool in the perinatal population is the BDI-2. Although this self-assessment tool for measuring depression symptoms was not originally intended for perinatal women, evidence has shown that the BDI is appropriate for use in this vulnerable population [45]. The BDI-2 total score ranges from 0–63. The scores were divided into four ranges, respectively: from 0 to 13 indicates a lack of depressive symptoms or their minimal occurrence, from 14 to 19 indicates mild depression, from 20 to 28 indicates moderate depression, and from 29 to 63 indicates severe depression [42].

#### 2.2.2. Maternal Anxiety Symptoms

We investigated two types of anxiety affecting the mental well-being of perinatal women during the pandemic: labour anxiety and COVID-19 pandemic-related anxiety. We used a dedicated Polish tool to measure the first area of anxiety under study—The self-report Labour Anxiety Questionnaire—LAQ (org. KwestionariuszLękuPorodowego KLP-II) [46]. This reliable and valid tool for measuring the intensity of labour anxiety in pregnant women consists of 9 items. The total score ranges from 0 to 27 and the ranges of results indicate four degrees of the symptom severity of labor anxiety (0 to 13—low, 14 to 15—slightly increased, 16 to 17—high, and ≥18—very high) [46]. In a multidisciplinary team of psychologists and psychiatrists, we also developed the COVID-19 Pandemic Anxiety Questionnaire (CRAQ) for self-reporting. The short 14 item survey refers to the concerns associated with the circumstances of the COVID-19 pandemic, the intensity of changes resulting from lockdown restrictions, and the participants’ attitudes towards these changes. The internal consistency of the total scale is satisfactory (Cronbach’s α = 0.79).

#### 2.2.3. Resilience

We measured the participant’s level of resilience through the Resilience Measure Questionnaire (KwestionariuszOcenyPrężności—KOP26) created by Gasior, Chodkiewicz, and Cechowski [47]. The questionnaire consists of 26 statements assessed on a five-point Likert scale, which refer to three areas of competence related to resilience: Family Competence (org. Kompetencje Rodzinne—KR), Personal Competence (Kompetencje Osobiste—KO), Social Competence (Kompetencje Społeczne—KS). The original Polish version of the questionnaire was used. The internal consistency in the original study is excellent for the total scale (Cronbach’s α = 0.90), satisfactory for subscales (KR = 0.90, KO = 0.82, KS = 0.78), and correlates well with other resilience scales (i.e., Ego Resiliency Scale in Kaczmarek’s adaptation ER/SPP, r = 0.59).

### 2.3. Data Analysis

We performed statistical analyses using STATISTICA 13.1 (TIBCO Software Inc., Palo Alto, CA, USA). Means, medians, and standard deviations were calculated for continuous variables. The categorical variables were summarized as frequency counts (counts and percentages). The normality of the distribution was verified using the Shapiro-Wilk test. Due to the division of participants into three groups depending on their level of resilience, we assessed the frequency differences between their demographic variables using a chi-square test. Associations between the variables were tested by Pearson’s correlation coefficient. Nonparametric Mann-Whitney and Kruskal-Wallis tests were used for subgroup comparisons that were unequal and/or did not meet the assumptions of homogeneity of variance. A Multiple Correspondence Analysis (MCA) was performed to detect the association between selected qualitative variables. The statistical significance level was set at *p* < 0.05. The a priori estimated sample size to meet the assumptions regarding the desired statistical power of the presented results was *N* = 85.

## 3. Results

### 3.1. Sample Characteristics

A total of 122 perinatal women met the inclusion criteria and provided complete questionnaires. Participants aged from 20 to 40 years old (M = 30.9; SD = 4.1), were predominantly in the third trimester of pregnancy (58.2%), primiparas (41.8%), with no current pregnancy complications (65.6%). The majority of the sample reported a higher educational status (88.5%) and lived in a large city (60.7%). A total of 98 participants (80.3%) reported no prior psychiatric treatment.

### 3.2. Resilience Level in the Study Sample

In our test sample, both the overall and partial resilience results were lower than in the Polish normative study [47]. For comparison, detailed descriptive statistics of resilience in our study population and normative study are presented in Table 2. Differences in the distribution of results obtained for our research sample in relation to the normalization sample concern the total result of the resilience score. In the normalization study, 30% of the study population scored in the low range (≤97), while in our study the percentage was almost double that (54%). Another noteworthy difference concerns the results in the Personal Competences subscale, in terms of the high level of results (≥40) in our population was almost five times lower (6,6%) than in the normalization sample (30%). The *t*-test for the difference of means was conducted confirming statistically significant differences between mean resilience and its components in 2016 (prepandemic) and study population. The total score of resilience (t = 4.53, *p* < 0.001) as well as scores of the following subscales: family relations (t = 2.49, *p* = 0.004), personal competence (t = 6.80, *p* < 0.001), social competence (t = 3.03, *p* = 0.0003) were significantly higher before the pandemic. Figure 1 shows the levels of resilience in the study group for the overall score and subscales. We divided the study group according to the overall resilience score according to the norms developed in the standardization study, where low scores are ≤97, medium scores are between 98 and 109, and high scores are ≥110 [47]. Detailed socio-demographic characteristics of our research sample in relation to the reported level of resilience are presented in Table 3. The chi-square analysis revealed no significant differences in terms of analyzed socio-demographic variables between subgroups.

### 3.3. Level of Depressiveness and Anxiety in the Study Sample

The general mean score on EPDS in the sample was 9 ± 5.5. Approximately 26.2% (*n* = 32) of the study sample had an EPDS score above the cut-off level of 13, indicating clinically significant levels of perinatal depressive symptoms. According to the BDI-2, 13.9% of participants reported moderate to severe symptoms of depression (scores ≥ 20). In addition, 15.6% of the women in the study had mild depressive symptoms based on the BDI-2 (scores 14–19). The mean labour anxiety total score obtained in the study sample was 14.4 ± 5.4, which oscillates in the middle of the score scale. Almost 40.1% of the participants achieved an increased score (≥16) on the labour anxiety scale, showing high or very high symptoms of anxiety related to childbirth. The main concern of women in the perinatal period related to the COVID-19 pandemic was separation from the newborn (mean score 8.1 out of 10). Separation from a close relative was the second highest-rated concern, scoring above the 70th percentile. Surprisingly, fear of COVID-19 infection or potential newbornCOVID-19-related harm was rated much lower.

### 3.4. Relationship between Depressiveness, Labour Anxiety, COVID-Related Anxiety and Resilience

We used Spearman’s correlation coefficients to identify associations between the variables (Table 4). We divided our research group into two subgroups in terms of the severity of depressive symptoms, measured using two tools: EPDS and BDI-2. In the case of BDI-2, we adopted the following ranges: scores from 14 to 19 for the first group and from 20 to 63 for the second analyzed group. Accordingly, for EPDS we used the following score ranges: 9–13 and 14–30. Moderate to very strong negative correlations were noted for overall resilience and its components. The strongest relationship was found between moderate to severe perinatal depression and resilience subscale of Personal Competence (r = −0.80; *p* < 0.05). A negative, though weak, correlationwas found between labour anxiety and resilience. We found no association between COVID-19 anxiety and resilience or any other measured variable. We also performed a multiple correspondence analysis (MCA) to detect the relationship between resilience, labour anxiety, and depressiveness in this sample of perinatal women. Very high and high perinatal anxiety along with lowest level of resilience clustered with EPDS and BDI-2 scores indicating depression (points representing depression are placed near points representing low resilience and labour anxiety). Two dimensions presented in Figure 2 clarify approximately 45% of inertia related to analyzed data.

## 4. Discussion

In this cohort of Polish women, which is part of a prospective, longitudinal study assessing the impact of the pandemic on the mental health of women in the perinatal period, we identified significant associations between low resilience and reduced mental well-being (depressive symptoms and labour anxiety). The research conducted so far indicates a high variability in the impact of COVID-19 lockdowns on women’s perinatal mental health [32]. Perhaps this variability may be reflected in the findings of a recent review on the trajectories of resilience and mental distress during the pandemic. The results suggest variability in resilience responses across the lifespan, indicating more resilient responses in older adults, but trajectories are less consistent for younger and older age groups compared with middle-aged adults [35]. In our research, approximately one-third of the women in the study population reported depressive symptoms. When making comparisons with the Polish population, although these rates are higher than before the pandemic [48], our study identified lower rates of depression than Studniczek and Kossakowska at an earlier stage of the pandemic [49]. This disparity may be due to two reasons. First, the cut-off value for depressive symptoms in the compared study was lower (11 or higher). Second, our study was conducted at a later stage of the pandemic, when knowledge about the risks of COVID-19 increased, vaccinations became more accessible, and obstetrics departments made earlier restrictions more flexible. Interestingly, similar results to the research mentioned above were obtained in our study population for the level of anxiety related to childbirth, with approximately 60% of the perinatal population achieving at least moderate results. Moreover, 26.2% of women reported a severe fear of childbirth, which is twice as high as the worldwide prevalence of tocophobia estimated at 14% [50].This percentage also exceeds those obtained in populations other than Polish during the pandemic, for example, 11.4% in the Netherlands population [51] or 4% in the population of China [52]. This alarming data may be related to the difficult situation of women in the perinatal period, resulting from restrictions related to the COVID-19 pandemic. Pregnant Polish women reported various problems in perinatal care, i.e., teleconsultations instead of standard visits, cancellations, or extended waiting time for visits. In the national survey, every fourth respondent (26%) resigned from some visits to the person conducting the pregnancy, every tenth—from some laboratory tests or CTG tests. Furthermore, 62% of pregnant women gave birth without the presence of a close relative, compared with 12% in 2018. For 72% of caesarean deliveries, there was no skin-to-skin contact between the newborn and a close relative [30].

The theoretical foundations of resilience assume the interaction between challenging events and the individual’s readiness to face them, which is the result of multiple factors, and is also modifiable [53,54]. Resilience has previously been identified as one of the factors that may impact perinatal mental health during the COVID-19 pandemic [32]. Higher levels of resilience in the perinatal population were associated with fewer depressive symptoms [55], and conversely, low levels of resilience were associated with poorer mental health outcomes [56]. In our study, we obtained significant negative correlations between resilience and depressiveness, with a correlation strength identical to the results obtained for the Italian population (Lombard group: r = −0.46) [55] and similar to the results of the Spanish study (r = −0.491) [57]. As previously established, resilience has been shown to have a protective effect on maternal mental health in facing adversity, trauma, and stress [36,38,58,59,60]. Therefore, implementing interventions that enhance the resilience of expectant parents could alleviate the devastating impact of mental disorders on maternal and child well-being. There is emerging evidence that psychoeducation, mindfulness-based interventions, and cognitive behavioral therapy-based interventions show promise in improving resilience factors [61].

In our study we also sought to determine whether factors related to the individual’s resilience can differentiate the response to stressful circumstances related to the pandemic, however, we found no association between COVID-19-related anxiety and resilience or any other measured variable. Therefore, we could not verify whether resilience has a mediating effect between COVID-19-related anxiety and depression and labour anxiety. Our findings are in contrast to Polish studies conducted during the pandemic that have investigated the mediating effect of resilience between pandemic stress and depressive symptoms [49]. Indeed, we used a different approach to measure stress related to the COVID-19 pandemic, which may be one of the reasons for this discrepancy. At the study design stage, no validated tool was available that would precisely address concerns related to the COVID-19 pandemic, therefore we decided to construct our measurement tool, achieving satisfactory reliability (Cronbach’s α = 0.79). It is also worth mentioning that Studniczek’s and Kossakowska’s study found only partial differences in the experience of prenatal stress related to the COVID-19 pandemic depending on the level of resilience and failed to confirm group differences in terms of infection stress and positive appraisal [49]. On the other hand, our findings are consistent with the results of Di Paolo’s study, in which no significant interactions between subjective distress and resilience were confirmed [60].The results of our study can also be understood in light of the theoretical basis of resilience. The concept of resilience assumes the flexible use of an individual’s resources in facing adversity [62]. The COVID-19 pandemic, with its medical [63], psychological [64], social, and economic [65] implications, can be seen as a global burden that has changed the lives of millions of people around the world, and the accompanying stress seems to be a universal social phenomenon, resilience is what differentiates mental health outcomes. Solid research shows that resilience is a key variable in mitigating and preventing the negative mental health consequences of the COVID-19 pandemic for individuals across populations [66]. This is also clearly shown by the results of our study.

Given the devastating impact of perinatal depression on the overall well-being of both mother and newborn, preventive psychological interventions are essential. The results obtained in our study have implications for obstetric care. First, our findings suggest that measuring resilience may be a useful routine in determining susceptibility to developing depression among women in the perinatal period. Psychological interventions to enhance resilience factors may have beneficial effects on mental health in this vulnerable population [37]. Third-generation behavioral therapies have been identified as effective in enhancing well-being through the use of resilience factors including mindfulness [67,68,69], acceptance [70], coping [71], and self-esteem [72].

Overall, our research is a solid contribution to understanding the impact of the COVID-19 pandemic on women’s mental well-being in the perinatal period. By employing reliable and well-validated measurement tools, we were able to ensure the accuracy and precision of our data, thereby enhancing the validity and robustness of our findings. However, some limitations of our research should be considered. Due to the cross-sectional design of the study, we cannot draw firm conclusions as to the direction of the causal relationship between the variables studied. Although every effort has been made to recruit a diverse and representative sample of participants, the limited size of our sample may affect the generalizability of our results. The relative homogeneity of our study population represents a notable limitation, particularly in light of existing research suggesting that socioeconomic factors likeyounger age, lower household income, lower education, unemployment, unmarried/unpartnered status, and a higher number of medical comorbidities are risk factors for low resilience [73]. Finally, the ethnicity of the sample, which was exclusively Polish, which minimizes the generalizability of the results to populations in other countries. 

To summarize, our study provides evidence that lower levels of resilience during pregnancy may be a significant predictor of increased severity of depressive symptoms and higher levels of anxiety related to childbirth among the perinatal population. These findings highlight the importance of considering resilience as an important factor in understanding and managing perinatal depression and may have implications for the development of targeted interventions. Disturbing data on the increase in the incidence of depressive and anxiety symptoms during the pandemic among this vulnerable population prompt preventive measures to protect mental well-being. This study adds to the growing body of perinatal mental health research during the COVID-19 pandemic and suggests that interventions to boost resilience may be an important avenue to reduce the burden of perinatal depression and anxiety.

## Figures and Tables

**Figure 1 jcm-12-06279-f001:**
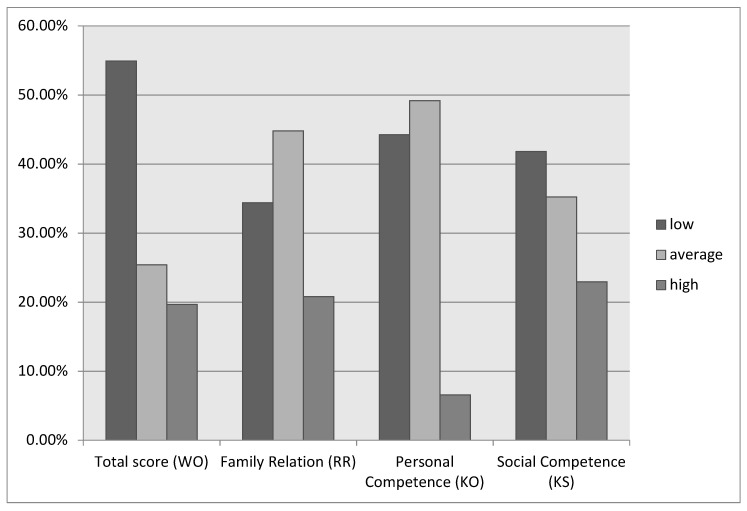
Resilience Levels in the Study Sample.

**Figure 2 jcm-12-06279-f002:**
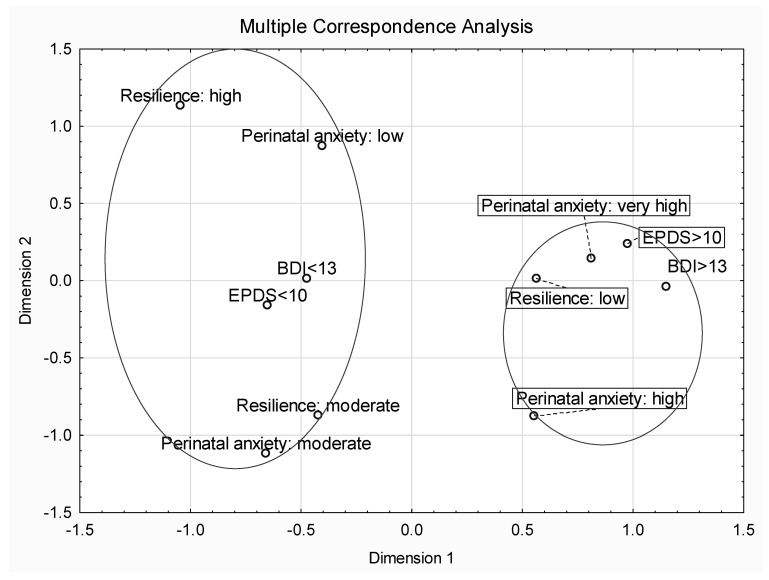
Multiple Correspondence Analysis (MCA) of resilience, depressiveness, and labour anxiety.

**Table 1 jcm-12-06279-t001:** Recommendations regarding perinatal care standards resulting from the pandemic.

The Issue of Perinatal Care	Recommendations Resulting from the Pandemic
Family birth	It is possible to limit the presence of a person accompanying childbirth in the event of an epidemic threat. The decision to enable a family delivery may be based on local organizational considerations, allowing for the isolation of women giving birth and their accompanying persons from other patients. Detailed requirements in the case of family deliveries should also be determined based on the capabilities of the maternity ward, and the necessary minimum should include, among others: the woman giving birth and her accompanying person stay in a single, individual delivery room equipped with a separate sanitary facility, the person accompanying the labour may be present from the beginning of labor and should leave the ward within 2 h after delivery, person in quarantine or isolation cannot participate in childbirth or enter the hospital premises.
Delivery of pregnant women infected with/suspected of COVID	Caesarean section is not indicated as the only method of delivery in patients diagnosed/suspected of COVID-19. The decision on the type of delivery is to be made depending on the current obstetric situation, taking into account local organizational conditions related to the state of the epidemic.
Skin-to-skin contact after a cesarean section	Skin-to-skin contact between the father and the baby after a cesarean section is possible while maintaining the necessary minimum, which should include: staying in a place designated by hospital staff where there is no contact with other patients or accompanying persons covering mouth and nose with a mask and periodically disinfecting hands throughout the stay in the hospital The decision on the possibility of skin-to-skin contact is to be made depending on the organizational conditions of the maternity ward.

**Table 2 jcm-12-06279-t002:** Descriptive statistics of resilience in the study population (*n* = 122) vs. normative study.

	Study Sample		[41]	
*KOP-26*	*M*	*SD*	*M*	*SD*
Family Relation (RR)	44.36	8.79	46.46	7.26
Personal Competence (KO)	30.90	7.03	35.43	5.27
Social Competence (KS)	18.57	5.47	20.15	4.27
Total score (WO)	94.06	18.18	102.04	12.98

**Table 3 jcm-12-06279-t003:** Socio-demographic characteristics of perinatal women in relation to the level of resilience.

Variable	Resilience Low		Resilience Average		Resilience High		X^2^	*p*
Age	M = 30	SD = 4.5	M = 32	SD = 3.2	M = 31	SD = 3.4		
Education							0.964	0.618
Higher education	*N* = 60	89.5%	*N* = 26	83.9%	*N* = 22	91.7%		
Secondary education	*N* = 7	10.5%	*N* = 5	16.1%	*N* = 2	8.3%		
Place of residence							1.986	0.738
Rural	*N* = 15	22.4%	*N* = 4	12.9%	*N* = 5	20.8%		
City up to 100,000 residents	*N* = 11	16.4%	*N* = 8	25.8%	*N* = 5	20.8%		
City with over 100,000 residents	*N* = 41	61.2%	*N* = 19	61.3%	*N* = 14	58.3%		
Current pregnancy status							0.855	0.652
Healthy	*N* = 46	68.7%	*N* = 20	64.5%	*N* = 14	58.3%		
Pregnancy Complications	*N* = 21	31.3%	*N* = 11	35.5%	*N* = 10	41.7%		
Pregnancy							3.801	0.434
First	*N* = 30	44.8%	*N* = 11	35.5%	*N* = 10	41.7%		
Second	*N* = 21	31.3%	*N* = 13	41.9%	*N* = 5	20.8%		
Third and subsequent	*N* = 16	23.9%	*N* = 7	22.6	*N* = 9	37.5		
Previous diagnosis of mental disorders							1.096	0.578
No	*N* = 52	77.6%	*N* = 25	80.6%	*N* = 21	87.5%		
Yes	*N* = 15	22.4%	*N* = 6	19.4%	*N* = 3	12.5%		
Previous COVID-19 infection							0.030	0.985
No	*N* = 52	77.6%	*N* = 24	77.4%	*N* = 19	79.2%		
Yes	*N* = 15	22.4%	*N* = 7	22.6%	*N* = 5	20.8%		

**Table 4 jcm-12-06279-t004:** Correlation matrix for tested variables.

Variable	1	2	3	4	5	6	7	8	9	10
1. Perinatal depression BDI–II 1	1									
2. Perinatal depression EPDS 1	0.17	1								
3. Perinatal depression BDI–II 2	−0.56	0.99	1							
4. Perinatal depression EPDS 2	0.34	**0.76**	0.63	1						
5. CRAQ	0.01	0.19	0.30	−0.26	1					
6. LA	**0.39**	−0.11	0.20	−0.08	0.14	1				
7. Resilience Total score (WO)	**−0.39**	**−0.46**	−0.64	−0.36	−0.07	**−0.21**	1			
8. Resilience Social Competence (KS)	**−0.34**	**−0.29**	−0.38	−0.31	−0.04	**−0.18**	**0.66**	1		
9. Resilience Family Relations (RR)	−0.24	**−0.41**	−0.64	−0.25	−0.03	−0.09	**0.90**	**0.34**	1	
10. Resilience Personal Competence (KO)	**−0.43**	**−0.47**	**−0.80**	−0.37	−0.11	**−0.29**	**0.93**	**0.50**	**0.81**	1

Marked tests are significant at *p* < 0.05.

## Data Availability

The data presented in this study are available on request from the corresponding author.

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
