# Peer review of "Resilience and Psychological Well-Being of Polish Women in the Perinatal Period during the COVID-19 Pandemic"

_jcm, 2023, doi:10.3390/jcm12196279_

Round 1
Reviewer 1 Report
This is a very concise study which shows promise to contribute novel findings to the literature base. Several recommendations are made below to improve the quality and transparency of the current manuscript.
Introduction
Very well constructed with inclusion of much pertinent perinatal literature. Though, given that you are looking at resilience and psychological wellbeing I was surprised to see so much content on depressive symptomatology and distress in your introduction. This section needs to be much condensed, and perhaps bring in literature on anxiety (particularly birth related anxiety) earlier on which is also prevalent postnatally. Generally speaking, though, there needs to be more of a positive psychology lens on the introduction to warrant the title of the study. There is perinatal resilience and wellbeing literature out there which you could bring in to substantiate your argument in the introduction.
In your introduction it would be useful to include a definition of resilience and summarise the literature base on resilience and psychological wellbeing in relation to pregnancy and the postpartum pre- and during- the pandemic.
Page 2 Line 69 – After talking about the global announcement of COVID-19 as being of international concern it would be good to include a brief summary of legislative changes and social distancing policies in Poland, perhaps including a table indicating key changes to provide a bit of context for the reader.
Line 27 about the pandemic being the largest threat of the 21st century needs referencing.
Methods
Did you calculate an a-priori sample size? This needs to be reported so we can have an idea of whether enough participants were recruited, as 122 currently seems quite and it raises concerns about the study being underpowered.
Results
Were there any attempts to control for psychiatric treatment as a potential confounder in analysis?
Page 3 Line 117 Include a justification for recruiting from Feb – Oct 2021 and give an overview of social distancing restrictions (or lack thereof) in Poland at that time.
Can you run some basic inferentials to investigate the difference between pre-pandemic and pandemic- scores of resilience? Other perinatal papers e.g., Fallon et al. (2021), have used pre-pandemic data to see change, recurrently.
Discussion
Page 9 When talking about the discrepancies in perinatal resilience literature it would be useful to discuss why these differences might exist.
Line 306 When talking about the protective effects of resilience against depressive symptomatology, discuss potential implications. What can be done with this information for policy? Practice? Research?
Line 317 Expand on why the use of a different measure might have contributed to different findings, and what we can do with this information.
Author Response
Dear Reviewer,
We are grateful for the opportunity to resubmit the revised version of manuscript for this special issue and your appreciation of our work. We are deeply thankful for your efforts and time spent reviewing our article to make it more comprehensive and valuable. We sincerely thank for your in-depth review of our manuscript and for all the thoughtful suggestions we received. We have made a concerted effort to respond appropriately to each suggestion we receive.
We have submitted the revised manuscript. Below we provide the point-by-point responses. All modifications in the manuscript have been highlighted in red.
We believe our revised manuscript addresses your suggestions and that the following fruitful changes have improved the revised manuscript for publication.
Comments from the Reviewer:
This is a very concise study which shows promise to contribute novel findings to the literature base. Several recommendations are made below to improve the quality and transparency of the current manuscript.
Point 1: Very well constructed with inclusion of much pertinent perinatal literature. Though, given that you are looking at resilience and psychological wellbeing I was surprised to see so much content on depressive symptomatology and distress in your introduction. This section needs to be much condensed, and perhaps bring in literature on anxiety (particularly birth related anxiety) earlier on which is also prevalent postnatally. Generally speaking, though, there needs to be more of a positive psychology lens on the introduction to warrant the title of the study. There is perinatal resilience and wellbeing literature out there which you could bring in to substantiate your argument in the introduction.
Response 1: We are grateful for your appreciation of our efforts to provide a coherent background to our research. Despite these efforts, the introduction included a disproportionate amount of information regarding the variables we investigated, which we corrected thanks to your valuable suggestions. We reviewed the literature on anxiety in the perinatal period and placed greater emphasis on the resilience factors of the perinatal population identified so far. Changes were made in the first and fifth paragraphs of the introduction.
Point 2: In your introduction it would be useful to include a definition of resilience and summaries the literature base on resilience and psychological wellbeing in relation to pregnancy and the postpartum pre- and during- the pandemic.
Response 2: Thank you for this suggestion. We expanded the definition of resilience and summarized the literature on resilience and mental well-being in the perinatal population before and during the pandemic. The changes apply to the fifth paragraph of the introduction.
Point 3: Page 2 Line 69 – After talking about the global announcement of COVID-19 as being of international concern it would be good to include a brief summary of legislative changes and social distancing policies in Poland, perhaps including a table indicating key changes to provide a bit of context for the reader.
Response 3: Thank you for this valuable feedback. We have created a table containing recommendations regarding the standards of perinatal care that emerged as a result of the pandemic during the third and fourth waves of the epidemic
(Table 1, Pg 3).
Point 4: Line 27 about the pandemic being the largest threat of the 21st century needs referencing.
Response 4: We have rephrased this statement and added a literature reference.
(Pg 2, Ln 39)
Point 5: Did you calculate an a-priori sample size? This needs to be reported so we can have an idea of whether enough participants were recruited, as 122 currently seems quite and it raises concerns about the study being underpowered.
Response 5: Thank you for your thoughtful suggestion. We understand the importance of an appropriate sample size to meet the assumptions regarding the desired statistical power of the presented results. Using online tool, https://sample-size.net/correlation-sample-size/ (access date 1.01.2021), we calculated required sample size for correlation coefficient. To detect a simple correlation r (r=0.3) of N observations, using a two sided test, 5% significance level test (α=0.05) with power 80% power (β=0.2), the estimated sample size is approximate 85 (n=85).
Point 6: Were there any attempts to control for psychiatric treatment as a potential confounder in analysis?
Response 6: Thank you for this inspiring question. In our study, we controlled whether the woman was taking pharmacological treatment or not, without obtaining detailed information about the medications implemented. We performed analyzes comparing the correlation between EPDS and BDI scores and resilience scores in treated and untreated perinatal women and found no statistically significant differences between these groups.
Point 7: Page 3 Line 117 Include a justification for recruiting from Feb – Oct 2021 and give an overview of social distancing restrictions (or lack thereof) in Poland at that time.
Response 7: In Study Population and Methods section (Pg 4) we have justified the time frame of the recruitment process by referring to changes in restrictions and limitations related to the pandemic.
Point 8: Can you run some basic inferentials to investigate the difference between pre-pandemic and pandemic- scores of resilience? Other perinatal papers e.g., Fallon et al. (2021), have used pre-pandemic data to see change, recurrently.
Response 8: Thank you for attention to this matter. In Resilience Level in the Study Sample section (Pg 6), we added information about differences in results before and during the pandemic, but we did not find any studies on the perinatal population that used the same measurement tools as in our study.
Point 9: Page 9 When talking about the discrepancies in perinatal resilience literature it would be useful to discuss why these differences might exist.
Response 9: We appreciate your valuable suggestion. The issue of high variability in the impact of COVID-19 restrictions on women's perinatal mental health seems important for understanding resilience factors. This was partly the focus of a recent systematic review of the trajectories of resilience and psychological distress during the pandemic, the results of which we briefly discussed
(Pg 10, Ln 3-7)
Point 10: Line 306 When talking about the protective effects of resilience against depressive symptomatology, discuss potential implications. What can be done with this information for policy? Practice? Research?
Response 10: Thank you for pointing out the implications of the discussed results that were not considered in our work. We have supplemented the manuscript with missing information
(Pg 10, Ln 42-46).
Point 11: Line 317 Expand on why the use of a different measure might have contributed to different findings, and what we can do with this information.
Response 11: We further discussed the reason for choosing this measurement and how our results relate to other findings.
(Pg 11, Ln 2-10)
We appreciate your input and have taken careful steps to address the concerns you raised, fostering greater clarity and precision in our work.
Best regards,
Authors
Reviewer 2 Report
The manuscript addresses depressive symptoms in the perinatal period among Polish women and the role that resilience might have in buffering negative mental health outcomes during COVID-19.
The manuscript is well-articulated and the literature quoted seems appropriate to me. However, there are some minor issues that I describe below:
· Line 46: what does “furthermore period prevalence rates” mean?
· Lines 76-77: The authors state, “The pandemic period is the most serious threat of the 21st century affecting the mental health of pregnant women around the world.” This sentence sounds a little pretentious to me, since in countries at war, for instance in the Third World - but not only, COVID-19 might have been perceived as less relevant than the person’s actual survival. Furthermore, the authors do not quote any relevant piece of literature to justify the expression “around the world”. For instance, the research by Scandurra et al. on perinatal depression addresses precisely the role of loneliness, anxiety, and maternal support during COVID-19 in a well-developed country such as Italy. Overall, the authors should also check the literature from other countries to conclude that COVID-19 was “the most serious threat of the 21st century affecting the mental health of pregnant women”.
· Lines 126-128: The authors state, “we sent participants an e-mail with information about the threat to their mental health and indicated possible sources of psychological and medical help”. Can the authors better explain what kind of information they provided to these women and in what consisted the possible sources of help?
· In the Methodology section, the authors do not mention the ethnicity of the sample. I guess it is mainly Polish, but if it is so, in the Limitations section it should be noted as a limitation.
I would encourage the authors to revise the manuscript as it contains several typos and the English is not always fluent.
Author Response
Dear Reviewer,
We are grateful for the opportunity to resubmit the revised version of manuscript for this special issue. We deeply appreciate your efforts and time spent reviewing our article to make it more comprehensive and valuable. We sincerely thank for your in-depth review of our manuscript and for the thoughtful suggestions we have received. We have made concerted efforts to respond appropriately to every suggestion we receive.
We have submitted the revised manuscript. Below we provide the point-by-point responses. All modifications in the manuscript have been highlighted in red.
We believe our revised manuscript addresses your suggestions and that the following fruitful changes have improved the revised manuscript for publication.
Comments from the Reviewer:
The manuscript addresses depressive symptoms in the perinatal period among Polish women and the role that resilience might have in buffering negative mental health outcomes during COVID-19.
The manuscript is well-articulated and the literature quoted seems appropriate to me. However, there are some minor issues that I describe below:
Point 1: Line 46: what does “furthermore period prevalence rates” mean?
Response 1: Thank you for your careful interest. The period prevalence rates mentioned in our manuscript is the percentage of the population that has revealed depressive symptoms at any time during the antenatal and postnatal period. However, due to a reviewer's suggestion, the introduction has been reformulated so that the mentioned period prevalence is not included in the Introduction.
Point 2: Lines 76-77: The authors state, “The pandemic period is the most serious threat of the 21st century affecting the mental health of pregnant women around the world.” This sentence sounds a little pretentious to me, since in countries at war, for instance in the Third World - but not only, COVID-19 might have been perceived as less relevant than the person’s actual survival. Furthermore, the authors do not quote any relevant piece of literature to justify the expression “around the world”. For instance, the research by Scandurra et al. on perinatal depression addresses precisely the role of loneliness, anxiety, and maternal support during COVID-19 in a well-developed country such as Italy. Overall, the authors should also check the literature from other countries to conclude that COVID-19 was “the most serious threat of the 21st century affecting the mental health of pregnant women”.
Response 2: Thank you for such valuable feedback. We understand that the severity of the threat posed by the pandemic varied around the world, and we agree that other threats, such as armed conflict, may have had a stronger impact on women's well-being in different countries. We reformulated the quoted statements, referring to the literature.
(Pg 2, Ln 39)
Point 3: Lines 126-128: The authors state, “we sent participants an e-mail with information about the threat to their mental health and indicated possible sources of psychological and medical help”. Can the authors better explain what kind of information they provided to these women and in what consisted the possible sources of help?
Response 3: Thank you for your insightful interest. We have provided more information on this issue in Study Population and Methods section.
(Pg 5, Ln 2-6)
Point 4: In the Methodology section, the authors do not mention the ethnicity of the sample. I guess it is mainly Polish, but if it is so, in the Limitations section it should be noted as a limitation
Response 4: The Study Population and Methods section contains information about inclusion criteria (living in Poland, fluency in Polish to be able to answer survey questions) closely related to ethnic origin. Thank you for pointing out the limitations resulting from the ethnic homogeneity of the sample. We mentioned this in the Discussions.
(Pg 11, Ln 41-43)
We appreciate your input and have taken careful steps to address the concerns you raised, fostering greater clarity and precision in our work.
Best regards,
Authors